



**Orbital-scale climate dynamics impacts on Gzhelian peatland wildfire activity in**
**the Ordos Basin**
Wenxu Du[1], Dawei Lv[1*], Zhihui Zhang[1*], Munira Raji[2], Cuiyu Song[1], Luojing Wang[1], Zekuan Li[1],
Kai Cao[1], Ruoxiang Yuan[1], Yuzhuang Sun[3]
[1] Shandong Provincial Key Laboratory of Depositional Mineralization and Sedimentary Minerals,
College of Earth Sciences and Engineering, Shandong University of Science and Technology,
Qingdao 266590, China
[2] Sustainable Earth Institute, University of Plymouth, Devon, United Kingdom
[3] College of Earth Science and Engineering, Hebei University of Engineering, Handan 056038,
Heibei, China
*Corresponding author: College of Earth Sciences and Engineering, Shandong University of
Science and Technology, Qingdao 266590, China.
E-mail addresses: lvdawei95@163.com (D. Lv), zhzhihui@sdust.edu.cn (Z. Zhang).
**Abstract**
The Carboniferous, an important coal-forming period in geological history, was characterized
by extensive vegetation and high oxygen levels. Numerous wildfire evidence suggests that high
frequency of wildfire occurred at that time, specifically in peatlands. However, the control
mechanisms for changes in wildfire activity in peatlands during this period are still not clearly
understood. In this study, evidence from the Gzhelian in the Ordos Basin, such as the
inertinite/vitrinite (I/V) ratio, indicated the existence of different frequencies of wildfire activity at
that time. The CaO/MgO and CaO/MgO · $Al_2O_3$ climate indicators revealed that high-frequency
wildfires mainly occur in warm and humid climates. Based on former age constraints, we deduced
that orbital cycles (long eccentricity) controlled the climate influence on peatland wildfires during



the Gzhelian. When eccentricity was high, abundant sunshine and frequent rainfall led to warmer
and more humid peatlands. The latter environments were more favourable for vegetation
development, leading to increased fuel loads, which in turn led to more frequent wildfires.
Moreover, the Gzhelian global wildfire records, showed that evidence of wildfire during this period
was mainly located in areas with abundant tropical vegetation, supporting the view that wildfire
activity during this period was mainly controlled by the fuel loads. Although Hg could be produced
by peatland wildfires, but our results show that Hg was mainly from frequent volcanic activity
during this period.
**Keywords:** Wildfire; Inertinite; Gzhelian; Hg; Long eccentricity

## 33    1. Introduction

In recent years, information about wildfires preserved in coal seams has been exposed (Robson

et al., 2015; Hou et al., 2022; Zhang et al., 2022). The macerals in the pyrogenic inertinite (fossil
charcoal in coal), such as fusinite, semifusinite, and inertodetrinite, have been widely used as direct
evidence to demonstrate the occurrence of paleo-wildfires (Scott, 2010, 2022; Jasper et al., 2013;
Belcher and Hudspith, 2017; Uhl et al., 2022; Shen et al., 2023). The Carboniferous through
Permian were important coal-forming periods in geologic history where large deposits of coal
developed (Berner, 2003). Although wildfires of the period have been somewhat summarized by
previous authors (e.g., Scott, 2000; Glasspool and Scott, 2010; Glasspool et al. 2015; Jasper et al.
2021), there is a dearth of high-resolution wildfire records from the period. Glasspool et al. (2015)
concluded from the analysis of the inertinite in coal that the frequency of wildfires during the Late
Paleozoic was mainly due to the higher oxygen content in the atmosphere at that time. However,
more detailed controlled factors for Carboniferous through Permian wildfires need to be improved.

Many researchers have found that changes in charcoal abundance in sediments indicate

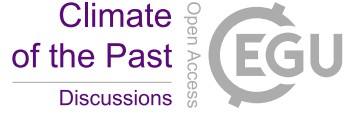

changes in orbital cycle forcing climate change, which in turn influences wildfire activity (Verardo
and Ruddiman, 1996; Thevenon et al., 2004; Hao et al., 2020; Shi et al., 2020; Cheng et al., 2022).
Daniau et al. (2013) found that the past 170 kyr of grassland wildfire activity in southern Africa has
been controlled by precession. Zhang et al. (2020a, 2023) found that wildfire activity during the
Early to Middle Jurassic was influenced by changes in long eccentricity (405 kyr) and the
precession cycle. The orbital period (eccentricity, obliquity, and precession) forces climate change,
thus affecting the amount and flammability of vegetation fuels to control wildfire activity (Zhou et
al., 2023). The study of whether wildfire activity in Carboniferous to Permian peatlands was
controlled by changes in orbital cycles is unclear. Late Carboniferous Gzhelian global wildfire
records indicate that the wildfire records were all distributed in tropical climate zones. Tropical
climate zones have abundant fuel accumulation and orbital cycle forcing wildfire activity may be
more pronounced (DiMichele, 2014).

As the largest Late Carboniferous to Early Permian peat basin in China, the Ordos Basin

contained a large amount of coal seams, which were preserved with numerous volcanic ashes and
wildfire products in the coal seams (Wang, 2017; Xu et al., 2020; Zhao et al., 2023; Zhang et al.,
2023a, 2023b). In this paper, the focus is on the No. 9 coal of the Yaogou Coal Mine in the Jungar
Coalfield of the Ordos torder to understand the occurrence of the Late Carboniferous wildfires and
the variation of burning temperatures. Employing the clear age constraints of the No.9 coal seam,
the possible role of long eccentricity orbital cycle changes in driving wildfires by means of wildfire
frequency and climate indicators at that time can be explored (Zhang et al., 2023a). Volcanic
activity, frequent wildfires, and other causes can lead to Hg enrichment, and this study also explores
the relationship between wildfire activity, volcanic activity, and Hg at that time, using changes in
Hg content in No. 9 coal.



## 2. Geological setting

The Ordos Basin, located at the western margin of the North China Craton, is the second largest terrestrial sedimentary basin in China (Ao et al., 2012; Zhang et al., 2023a). The Jungar Coalfield is in the southwestern part of the Inner Mongolia Autonomous Region of China and in the northeastern part of the Ordos Basin, which contains about 26.8 Gt of coal reserves, which is one of the richest coal reserves in northern China (Dai et al., 2006,2008,2012; Li et al., 2016). The coalfield contains coal-bearing sequences of the Carboniferous and Permian systems (Wang et al., 2011).

This study was conducted in the Yaogou Mine, in the northeastern part of the Jungar Coal Field (Fig. 1b). The coal-bearing sequences in the Yaogou Mine include the Benxi Formation, Taiyuan Formation, and Shanxi Formation, with a total thickness of 84~200 m (Zhang et al., 2023a). The Benxi Formation, with a total thickness of 84~200 m (Zhang et al., 2023a), lies unconformably on the Middle Ordovician Majiagou Formation, with a thickness of about 17~28 m. The Taiyuan Formation is underlying by the Benxi Formation. The Taiyuan Formation, with a thickness of about 31-75 m (average 58 m), conformably overlies the Benxi Formation. The Taiyuan Formation consists mainly of grey and gray-white quartz sandstone, siltstone, mudstone, and five coal beds marked from top to base as No. 6 to No. 10 (Fig. 1c) (Zhang et al., 2023a). Zhang et al. (2023a) determined by U-Pb zircon from altered volcanic ashes at the top and bottom of No. 9 coal that the Taiyuan Formation in this study area belongs to the Gzhelian stage of the Late Pennsylvanian. The Shanxi Formation with a thickness of about 35-97 m is mainly terrigenous coal-bearing clastic rocks, dominated by sandstones and coal seams (Zhang et al., 2023a).

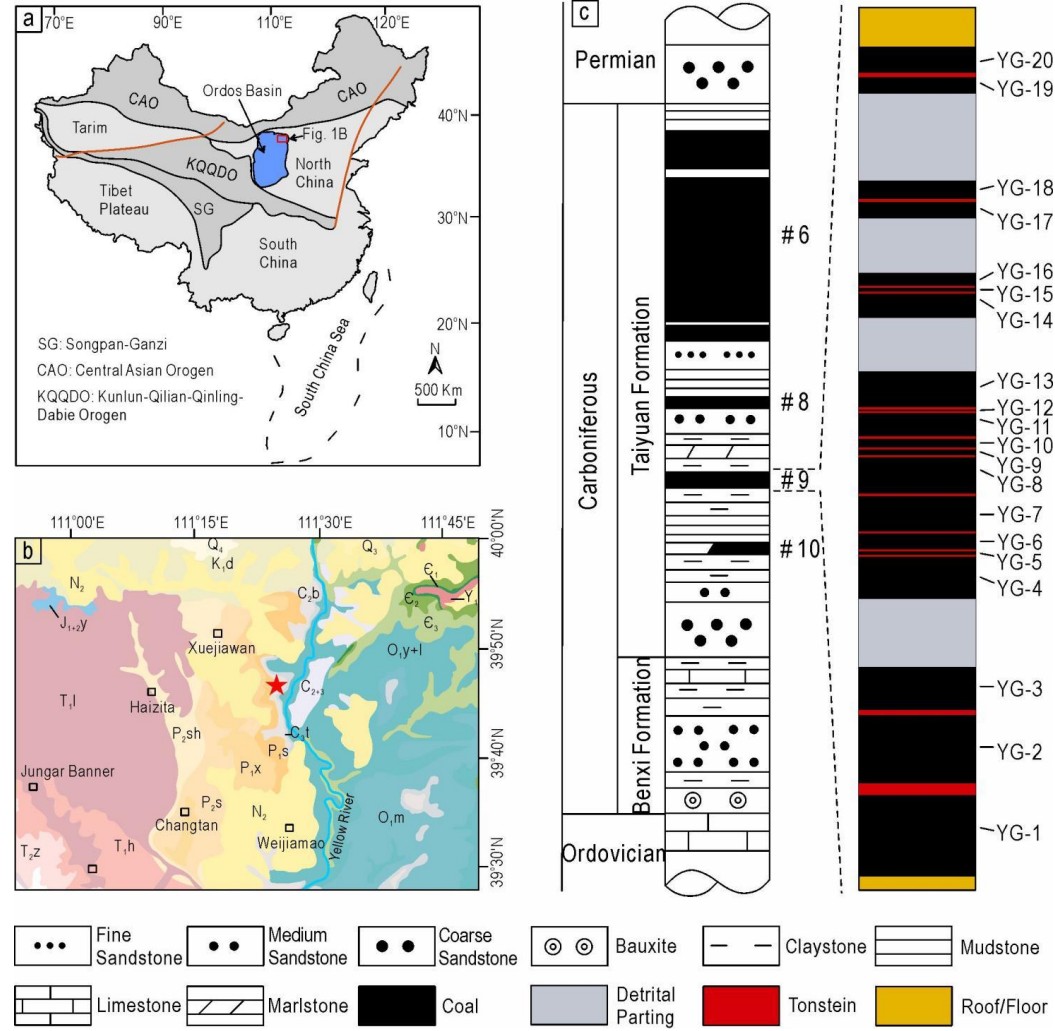

**Fig. 1** Comprehensive geologic map of the Yaogou coal mine in the Ordos Basin, northern China. (a) Map

showing the location of the Ordos Basin and the study area. (b) Geologic map of the study area showing the

location of the study area (red stars). (c) Sedimentary sequence of the Jungar Coal Field and columnar map of the

No. 9 coal seam. The No. 9 coal occurs in the Taiyuan Formation. $Q_4$-Holocene, $Q_3$-Pleistocene, $N_2$-Pliocene,

$K_1d$-Dongsheng Formation (Fm.), $J_{1+2}y$-Yan'an Fm., $T_2z$-Zhifang Fm., $T_1h$-Heshanggou Fm., $T_1l$-Liujiagou Fm.,

$P_2sh$-Shiqianfeng Fm., $P_2s$-upper Shihezi Fm., $P_1x$-lower Shihezi Fm., $P_1s$-Shanxi Fm., $C_3t$-Taiyuan Fm.,

$C_2b$-Benxi Fm., $O_1m$-Majiagou Fm., $O_1y+l$-Yeli and Liangjiashan Formations. Figure modified from Zhang et al.

(2023a).



## 3. Material and methods

### 3.1 Sampling

In this study, a total of 20 coal samples were collected from the exposed face of the No. 9 coal at the Yaogou Mine, with a cumulative thickness of the profile of about 6.1 m. To minimize contamination and oxidation, the samples were all immediately stored in plastic bags. From bottom to top, all coal samples were marked YG-1 to YG-20 (Fig. 1c).

### 3.2 Analytical method

The analytical methods used in this study to demonstrate wildfire characteristics included petrographic analysis, coal rock micro component identification, fusinite reflectance measurements, and scanning electron microscopy (SEM).

The collected fossil charcoal fragments were crushed to a < 20 top size and the crushed samples were embedded in epoxy resins. The reflectance of the fusinite was measured by the oil immersion at room temperature and determined using an MSP UV-VIS2000 spectrophotometer (Petersen and Lindström, 2012; Xu et al., 2020). To estimate the wildfire burning temperatures represented by the experimental samples, calculations were based on the correlation proposed by previous works:

$$T = 184.1 + 117.76 \times \%R_o, \tag{1}$$

with $R_o$ represents the reflectance of the inertinite (Jones, 1997; Petersen and Lindström, 2012). Based on the model of oxygen content proposed by Glasspool (2010, 2015), the oxygen content in the atmosphere was predicted for the period of this study:

$$I = (0.5 - 0.5\cos[\pi(O - O_{min})/(O_{max} - O_{min})])^n, \tag{2}$$

with I as the average inertinite content, O as the oxygen content, $O_{max}$ as the oxygen content when the inertinite content reached 100%, and $O_{min}$ as the oxygen content when was no inertinite.



The microstructure of fossilized charcoal samples was observed by SEM to make a clearer
observation of fossilized charcoal samples. The samples were observed under vacuum conditions
using a 10 kV accelerating voltage, standard beam light, and a secondary electron probe. All the
above experiments were completed at the Shandong Provincial Key Laboratory of Depositional
Mineralisation and Sedimentary Minerals, Shandong University of Science and Technology.
To determine the coal rock components, the coal samples taken were measured according to
the National Standard GB/T8899-2013. The reflectivity of the vitrinite group of the coal samples
was measured according to the National Standard GB/T 6948-2008. To improve the quantification
of charcoal abundance in each single sample, the ratio between inertinite maceral and vitrinite
macerals (I/V), was calculated to determine the frequency of wildfires (Zhang et al., 2022). All of
the latter experiments were completed in Xi'an Coal Science and Technology Ltd. All samples were
analyzed for Total Organic Carbon (TOC) according to the National Standard GB/T19145-2003. All
samples were analyzed for Hg concentration according to the National Standard GB/T
22105.1-2008. All above experiments were completed at the Beijing Qingchen Huanyu Petroleum
Geological Technology Co. Ltd.
*3.3 Data Synthesis*
To build a reliable database of the Gzhelian wildfire evidence, data was compiled from
peer-reviewed journal publications following the program of Lu (2021) and Lv (2024). Personal
data and unpublished materials were not included in our study. We used the keywords 'Gzhelian
wildfire', 'charcoal', 'inertinite', 'fusain', 'fusinite', and/or 'PAHs' to search papers in Google
Scholar, Web of Science, ScienceDirect, and JSTOR. Literature searches were conducted by the end
of April 2024.
To minimize the possibility of low credible/reliable wildfire data, the data published in the



original article was double-checked and evaluated before inclusion in our database. The evidence
for fossil charcoal in the database is based on the standardized guidelines in Scott (2000,2010,2020).
The distinction between sources of PAHs (pyrogenic and petrogenic) was made according to Yunker
et al. (2002), and only records of pyrogenic PAHs were included. For example, Hower et al. (2022)
had two inertinite data that did not have details of where samples were taken, so they were
discarded. The inertinite in Presswood et al. (2016) was due to magmatic intrusion, so it was
discarded.
Three types of Gzhelian wildfire evidence were included in the database: charcoal type I
(pyrogenic inertinite macerals from coals), charcoal type II (fossil charcoal from clastic sediments),
and pyrogenic PAHs. The location (e.g., country, state/province, city, latitude, longitude, etc.),
palaeogeographical localities (e.g., palaeocontinents, palaeolatitude, and palaeolongitude),
stratigraphic unit (e.g., formation, group), and rock type (e.g., coal, sandstone, mudstone) were
integrated for each wildfire record. When multiple studies reported wildfire data from the same
geologic unit in the same location (e.g., section, region, or basin), they were considered as one
record (Lu et al., 2021). Depending on the type of evidence, wildfire occurrences in the same study
site can be identified in different studies and recorded as one occurrence.
The GPlates 2.2.0 model constructed by Scotece (2008) was used to estimate paleolatitudes
and paleolongitudes data for wildfire occurrence. To provide a visual representation of the spatial
distribution, the occurrence of wildfires was projected on a paleogeographic map of Gzhelian by
Scotese (2021) (Fig. 9).
**4. Results**
*4.1 Coal rock micro component and geochemical characterization*
The microscopic component contents of the coal samples collected in the mineral-free state



from No.9 coal of Yaogou Mine are shown in Table 1. Of all the samples, the vitrinite was dominant,
with contents ranging from 64.5% to 84.7% (average = 73.5%). The inertinite content of the coal
samples ranged from 14.1% to 30.8%, with an average content of 22.3%. The liptinite had the
lowest content, ranging from 1.3% to 8.7%, with an average content of 4.2%. Nearly all inertinite
preserved in coal was defined as an incomplete combustion product, comparable to fossil charcoal
(Scott 2000, 2010; Scott and Glasspool 2006, 2007; Diessel, 2010). Because of this, subcomponent
analyses were analyzed for the inertinite in the coal samples Table 1. In the 20 coal samples, the
main component of the inertinite was semifusinite, with contents ranging from 7.4% to 17.2% with
an average content of 11.5%. The content of inertodedetrinite was rather low, ranging from 1.8% to
13.4%, with an average content of 6.6%. The contents of fusinite, macrinite, and micrinite were low,
with average contents of 1.2%, 2.3% and 0.7%. Among the 20 samples, the variation of
inertinite/vitrinite ranged from 0.17 to 0.47, with a mean value of 0.31. The reflectance ($R_{vo}$) of the
vitrinite of the 20 coal samples ranged from 0.37% to 0.66.

The TOC content of the 20 coal samples from the Taiyuan Formation ranged from 27.82% to

48.07%, with an average content of 39.49% (Table 2). The Hg content of the 20 samples varied
greatly, ranging from 29.5 ppb to 393 ppb, with an average of 134.96 ppb (Table 2). Zhang et al.
(2023a) analyzed the sulfur content, $Al_2O_3$ content, and ash yield of 20 coal seams in the Taiyuan
Formation in this study area (Table 2). The Spearman correlation analysis showed that the
correlation between Hg and $Al_2O_3$ in the 20 coal samples from Taiyuan Formation is highly
significant, with positive correlation, the correlation coefficient is +0.529, and the significance is
0.016 (Fig. 6a). The correlation between Hg and sulfur is not significant with the correlation
coefficient of -0.175 and significance of 0.46 (Fig. 6b). The correlation between Hg and TOC is
significant, with a correlation coefficient of -0.659, and the significance is 0.002 (Fig. 6c). Since the



source of elemental aluminum in coal is mainly clay minerals, the correlation between elemental Al
and ash yield was analyzed. The correlation between elemental Al and ash yield in the coal samples
of this study was highly significant with a correlation coefficient of +0.776 and significance of 0.
The correlation between elemental Hg and ash yield was significant and positive with the
correlation coefficient of +0.481 and significance of 0.032 (Fig. 6d).

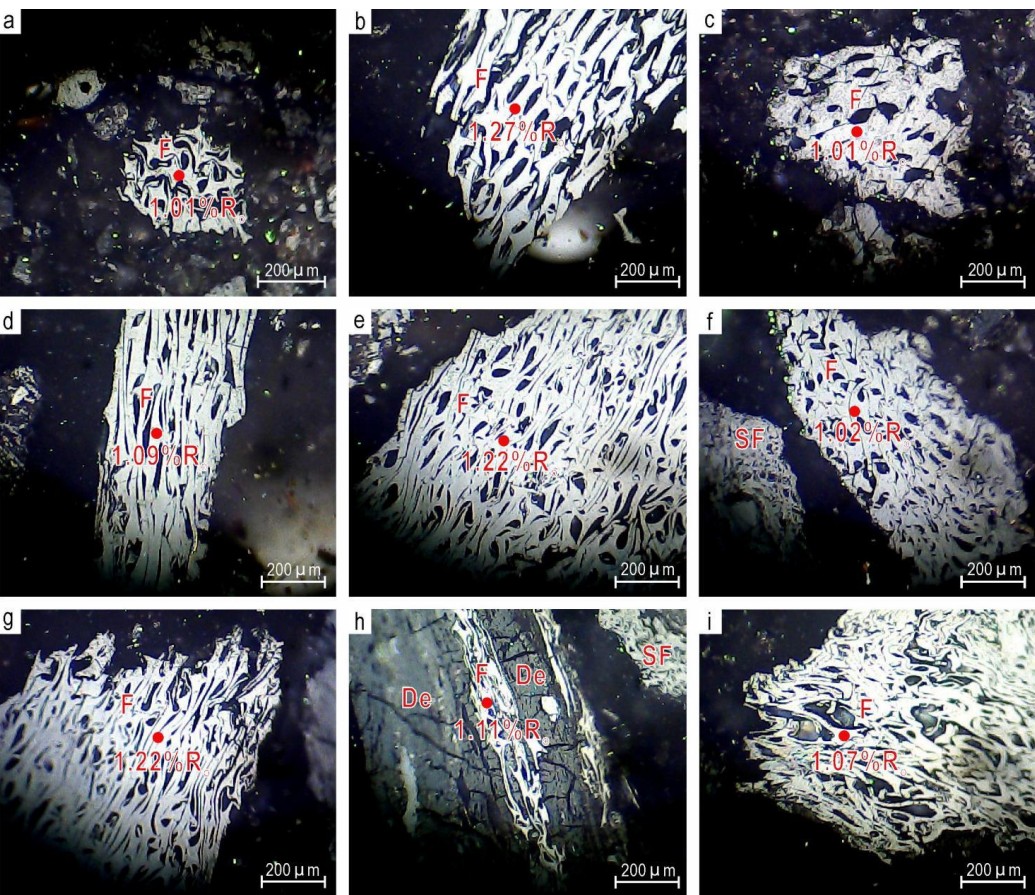


**Fig. 2** Features of fossil charcoal in the Ordos Basin Taiyuan Formation observed under the oil immersion optical

microscopy. F-fusinite; SF-semifusinite; Red circles show the test points, and numbers show the reflectance

values of the points.



**Table 1**
The coal micro component contents of 20 seams of No. 9 coal from Yaogou Mine in Ordos Basin

| Sample No. | Percentage of the total organic macerals (vol.%) | | | Percentage of the total inertinite macerals (vol.%) | | | | | TOM (vol.%) | Total minerals (vol.%) |
|---|---|---|---|---|---|---|---|---|---|---|
| | Vitrinite | Inertinite | Liptinite | Fusinite | Semifusinite | Macrinite | Micrinite | Inertodetrinite | | |
| YG-1 | 79.6 | 14.6 | 5.8 | 1.1 | 9.3 | 0 | 0.2 | 4 | 53.9 | 46.1 |
| YG-2 | 80.5 | 16 | 3.5 | 2.4 | 10.6 | 0.6 | 0.6 | 1.8 | 60.4 | 39.6 |
| YG-3 | 68.3 | 27.4 | 4.3 | 0.9 | 14.1 | 1.5 | 2.6 | 8.3 | 81.2 | 18.8 |
| YG-4 | 64.5 | 26.8 | 8.7 | 0.4 | 13.3 | 2.4 | 0.4 | 10.3 | 84.8 | 15.2 |
| YG-5 | 84.7 | 11.9 | 3.4 | 0.4 | 7.4 | 1.5 | 0.2 | 2.4 | 84.9 | 15.1 |
| YG-6 | 65.2 | 30.8 | 4 | 0 | 14.6 | 2.4 | 0.4 | 13.4 | 86.2 | 13.8 |
| YG-7 | 72.2 | 23 | 4.8 | 3 | 10.7 | 3 | 0.8 | 5.5 | 87.1 | 12.9 |
| YG-8 | 70.8 | 21.5 | 7.7 | 2.5 | 12.2 | 1.5 | 0.2 | 5.1 | 81.8 | 18.2 |
| YG-9 | 78.4 | 19.1 | 2.5 | 0 | 9.8 | 2.5 | 0.4 | 6.4 | 80.1 | 19.9 |
| YG-10 | 66.1 | 27.4 | 6.5 | 3.8 | 13.1 | 3.2 | 0.6 | 6.7 | 83.9 | 16.1 |
| YG-11 | 75.2 | 23.5 | 1.3 | 0.6 | 12 | 3.4 | 0.2 | 7.3 | 79.6 | 20.4 |
| YG-12 | 72.1 | 25.1 | 2.8 | 0.9 | 15.4 | 2.2 | 0.4 | 6.2 | 73.9 | 26.1 |
| YG-13 | 65.8 | 30.2 | 4 | 0.8 | 17.2 | 4.1 | 1.8 | 6.3 | 86.2 | 13.8 |
| YG-14 | 82 | 14.1 | 3.9 | 0 | 8.3 | 2.1 | 0 | 3.7 | 81.3 | 18.7 |
| YG-15 | 80.2 | 18.4 | 1.4 | 0 | 11.2 | 0.8 | 0.6 | 5.8 | 77 | 23 |
| YG-16 | 81.8 | 16.5 | 1.7 | 0 | 10.4 | 1.2 | 1.2 | 3.7 | 85.2 | 14.8 |
| YG-17 | 78.1 | 18.4 | 3.5 | 2.7 | 9 | 1.7 | 0.5 | 4.5 | 59.5 | 40.5 |
| YG-18 | 66.8 | 28.9 | 4.3 | 0.8 | 11.3 | 4.7 | 0 | 12.1 | 59.9 | 40.1 |
| YG-19 | 73 | 24.9 | 2.1 | 1.1 | 8.9 | 2.9 | 0.4 | 11.6 | 71.5 | 28.5 |
| YG-20 | 65 | 27.2 | 7.8 | 3.2 | 11 | 4.1 | 2 | 6.9 | 81.8 | 18.2 |






**Table 2**
The chemical element data of 20 coal seams from the No. 9 coal of the Yaogou coal mine in the Ordos Basin. The
$Al_2O_3$, TS, and TOC were from Zhang et al. (2023a).

| Sample | Hg (ppb) | $Al_2O_3$ (%) | TS (%) | TOC (%) |
|--------|----------|---------------|--------|---------|
| YG-1   | 255      | 16.33         | 0.23   | 27.82   |
| YG-2   | 145      | 11.94         | 0.39   | 29.92   |
| YG-3   | 63.2     | 7.67          | 0.44   | 47.30   |
| YG-4   | 118      | 9.46          | 0.77   | 44.14   |
| YG-5   | 58.6     | 8.96          | 0.32   | 48.07   |
| YG-6   | 70.4     | 12.75         | 0.41   | 42.04   |
| YG-7   | 85.1     | 7.44          | 0.57   | 44.72   |
| YG-8   | 35.4     | 7.42          | 0.52   | 48.04   |
| YG-9   | 43.5     | 11.87         | 0.45   | 41.17   |
| YG-10  | 269      | 10.5          | 0.59   | 41.51   |
| YG-11  | 88.5     | 10.53         | 0.48   | 38.10   |
| YG-12  | 83.9     | 10.74         | 0.5    | 35.33   |
| YG-13  | 34.5     | 9.57          | 0.48   | 46.26   |
| YG-14  | 156      | 12.77         | 0.35   | 40.37   |
| YG-15  | 352      | 16.48         | 0.42   | 32.35   |
| YG-16  | 237      | 14.23         | 0.32   | 36.72   |
| YG-17  | 393      | 12.42         | 0.36   | 29.26   |
| YG-18  | 152      | 17.88         | 0.29   | 32.05   |
| YG-19  | 29.5     | 13.53         | 0.3    | 37.95   |
| YG-20  | 29.6     | 7.57          | 0.44   | 46.68   |

*4.2 Microscopic observations of fossil charcoal*

Under the oil-immersion reflected-light microscope, a large amount of semifusinite was

observed in the coal samples of the Taiyuan Formation. Some representative micrographs are
provided in Fig. 2. The pore sizes of the semifusinite ranged from 40 - 120 μm , with a small
variation in pore size. The measured reflectance of the inertinite of 20 coal samples from the
Taiyuan Formation ranged from 1.01% to 2.07%, with the average value of 1.66%.

The structural features of the fibre structure and the cell wall in the fossil charcoal fragments

can be observed by SEM. The majorityt of the cellular structures were crushed or broken(Fig. 3a,c).
As shown by the radial structure, the tubular cells were straight and 10-20 μm wide (Fig. 3a). There
was one single row of circular or elliptical pitting with 0.5-2 μm width (Fig. 3b). And the cell walls
of the samples showed homogenization (Fig. 3d).

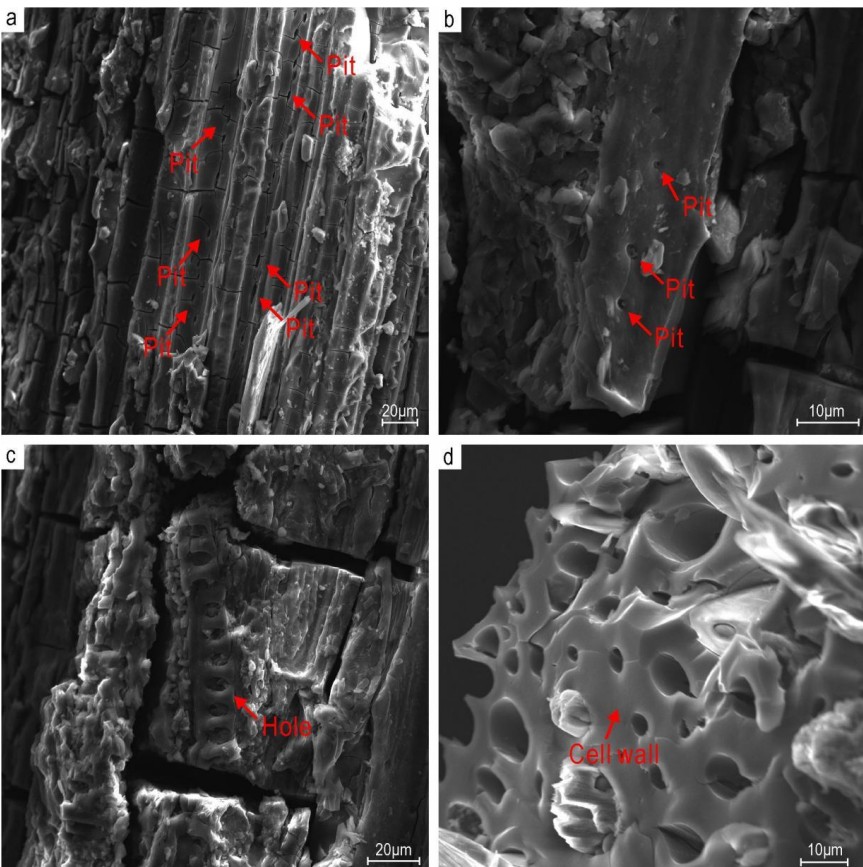


**Fig. 3** Scanning electron microscope electron probe image of fossilized charcoal. (a-b) Red arrows point to
uniseriate pits. (c) Red arrow points to larger holes in the cell wall that may represent pits, which were
diagenetically enlarged during charring. (d) Red arrow points to homogenized cell wall.
*4.3 The Gzhelian wildfire record in geologic context*

Published wildfire records for the Gzhelian were compiled in Table S1. Based on the type of

evidence these can be divided into five groups: 23 records of charcoal type I (79.4% of the total
records), two records of charcoal type II (6.9%), one record of pyrogenic PAHs (3.4%), two records
of charcoal type I and charcoal type II (6.9%), and one record of charcoal type II and pyrogenic
PAHs (3.4%). The published Gzhelian wildfire data were observed in 19 geologic units (or


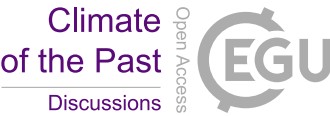

formations), with two record from an unclear unit.
All of the 29 reported Gzhelian wildfires have occurred in the lower latitudes (30°N-30°S) (Fig.
9). According to Boucot et al.'s (2013) suggested paleoclimatic classification, the Gzhelian can be
divided into four different paleoclimatic zones: cool temperate, warm temperate, arid and tropical.
The Gzhelian wildfires all occurred in the tropical area (Fig. 9).
**5. Discussion**
*5.1 Repeated wildfire in the Gzhelian of the Ordos Basin*
The origin of the inertinite in coal has been controversial, and it has been suggested that the
fusinite and semifusinite in the inertinite may be produced from fungal degradation of vegetation
tissues in oxidizing environments or influenced by other microbial activities (Beeston, 1987;
Teichmüller, 1989; Varma, 1996; ICCP, 2001). Some semifusinite in coal may also have been
formed during diagenesis (Hudspith and Belcher, 2020). Nevertheless, research suggests that the
fusinite and semifusinite in coal were equivalent to fossil charcoal (Scott and Glasspool, 2007;
Hudspith et al., 2012), which was produced by the incomplete combustion of vegetation by
wildfires (Scott, 2010; Liu et al., 2022). All coal samples from the Taiyuan Formation had low
levels of both macrinite and micrinite (Table 1). Both fusinite and semifusinite preserved complete
cellular structures (Fig. 2), suggesting that the fusinite and semifusinite in the study samples may be
products of vegetation combustion.
In addition, the low reflectance of the vitrinite in the coal samples, averaging 0.49%, indicated
that the samples were poorly metamorphosed, and was not enough to support the claim that the
semifusinite in the samples was due to diagenesis. Experiments had shown that the carbonization
temperature for homogenization of vegetation cell walls needs to be at least higher than 250 - 300
°C (Scott and Jones, 1991; Osterkamp et al., 2017). Obvious homogenization of the cell walls of the





fossil charcoal could be observed under the SEM (Fig. 3d), which more clearly suggested that the
fossil charcoal in the samples was caused by wildfire burning. The inertinite content of the Taiyuan
Formation coal samples showed a small range of variability (14.1% - 30.8%) and was dominated by
semifusinite (7.4% - 17.2%), which indicates the presence of frequent small wildfires during this
period. Under the SEM, it can be observed that the cellular structure of the fossil charcoal from the
Taiyuan Formation is obviously broken, and some cellular cavities in the charcoal appeared to have
different degrees of deformation (Fig. 3c), which may be due to the tectonic movement of the strata.
Scott (2010) divided wildfires into three types based on their strength: surface fire, ground fire,
and crown fire. Among these, surface fires whose fuels are mainly surface herbaceous, shrubs and
various biological wastes, burn at lower temperatures, usually below 400 ℃. Ground fires whose
fuel is mainly the dry humus layer in the soil, which can burn thick layers of peat, burn at relatively
high temperatures, up to 600 ℃. It has been demonstrated that the reflectance of the inertinite is
positively correlated with the combustion temperature, with higher inertinite reflectance indicating
higher combustion temperatures (Jones et al., 1991; Guo and Bustin, 1998; Scott and Glasspool,
2007; McParland et al., 2009). The reflectance of the inertinite in the Taiyuan Formation samples
varied in a small range (1.01%-2.07%), and it can be inferred that the Taiyuan Formation wildfire
burned at temperatures ranging was about 300 ℃ to 430 ℃, with an average burn temperature of
379 ℃ (after formula Scott and Glasspool, 2015). This suggests that mainly low-temperature fires
occurred in the Ordos Basin during the Gzhelian (Fig. 4). Wildfire burning may have been fueled
primarily by low-growing vegetation or biological waste, which would be consistent with the
homogenization of fern cell walls observed under the SEM.

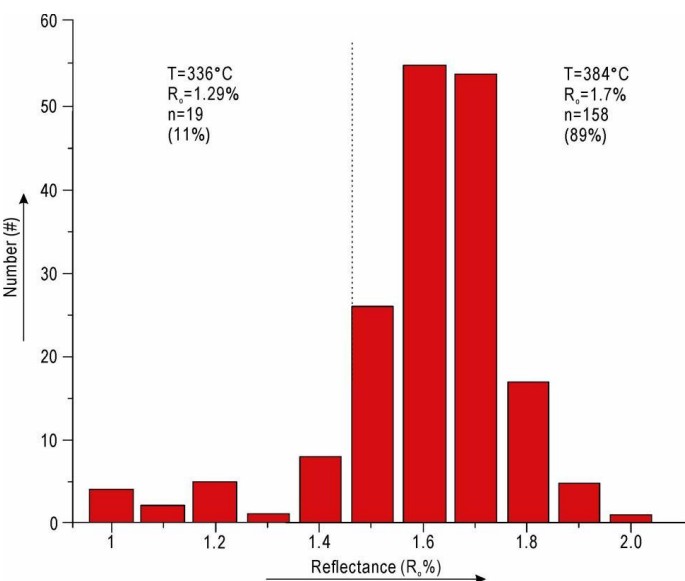

**Fig. 4** The inertinite reflectance histograms indicate calculated combustion temperatures (Shukla et al., 2023).

*5.2 Sources of elemental mercury in coal*

Soils are the largest global reservoir of Hg and forest ecosystems are one of the key areas in the Hg cycle (Wang et al., 2019). Peatland vegetation can retain atmospheric Hg in their bodies by absorbing it (Grigal, 2003), while humus and sulfides in peatlands have a high affinity for Hg, which makes them important sinks for Hg (Yudovich and Ketris, 2005; Woerndle et al., 2018). In the pre-anthropic era, the main sources of Hg in the surface environment were volcanic eruptions, intense continental weathering, and wildfires (Pavlish et al., 2003; Pyle and Mather, 2003; Selin, 2009; Shen et al., 2020).

In this study, Hg was negatively correlated with Total Organic Carbon (TOC) and Total Sulfur (TS) (Fig. 6), indicating that Hg is not predominantly present in organic matter and sulfides. Correlation of elemental concentration with ash yield may provide preliminary information on their affinity with organic and inorganic (Eskanazy et al., 2010; Kortenski and Sotirov, 2002; Dai et al., 2012a; Zhang et al., 2023a). Mercury has a relatively high correlation (r=0.481) with ash yield, indicating a dominant inorganic affinity. There was a high correlation between Hg and $Al_2O_3$ in this



study (r=0.529), indicating that Hg was mainly enriched in clay minerals (Fig. 6). The Hg

concentrations in coal samples YG-1, YG-10, YG-15, YG-16, and YG-17 (237-352 ppb) were

generally higher than the average concentration of coals from around the world (100 ppb) and the

average concentration of Chinese coals (163 ppb) (Dai et al., 2012b). After normalizing for $Al_2O_3$, it

still showed anomalously high values in these five coal samples, indicating that these five Hg peaks

are true Hg anomalies (Fig. 5) (Xie et al., 2022).

Xie et al. (2022) demonstrated that wildfire burning resulted in the release of most of the Hg

accumulated in vegetation and soils to the atmosphere, and that atmospheric Hg accumulated

quickly in peatlands, leading to Hg anomalies enrichment in peatlands. Comparing the Hg levels of

the samples in this study with the frequency of wildfires has not shown that the anomalously high

values of Hg are well correlated with the frequency of wildfires (Fig. 5). There were even

anomalously high values of Hg in samples YG-15, YG-16, and YG-17, but the frequency of

wildfires was significantly reduced. This indicates that the anomalous enrichment of Hg in this

study was not caused by wildfire burning.

The mass of Hg in forest vegetation is about 0.1 $mg/m^2$ , however, the mass of Hg in peatlands

is about 20 $mg/m^2$ , which is much greater than the amount in vegetation (Grigal, 2003; Turetsky et

al., 2006). The types of wildfire burning in Xie et al. (2022) were primarily ground and surface fires.

Ground fires were fueled primarily by a dry humus layer in the soil (Scott, 2010), which can burn a

thicker layer of peat, resulting in the release of Hg from the peatland. In this study, the type of

wildfire burning was predominantly surface fires, with the primary fuels for surface fires were low

surface vegetation and biological litter (Scott, 2010), which coincides with the observation of ferns

as fuels in the SEM (Fig. 3). Ku et al. (2018) found that different wildfire intensities had different

effects on Hg volatilization from vegetation. Therefore, wildfire burning may not necessarily lead to





Hg enrichment, and wildfire types may not have the same effect on Hg. Mercury enrichment is
favored when wildfires were primarily ground fires, where burning of the peat layer resulted in the
release of large quantities of Hg, which were absorbed by the peatland.

314        Excluding the anomalous enrichment of Hg in coal due to wildfire activities, the known causes

of the anomalous enrichment of Hg in coal are mainly the volcanic activities during the early
peatland accretionary phase that led to the increase of Hg input (Roos-Barraclough et al., 2002;
Yudovich and Ketris, 2005), and the invasion of low-temperature hydrothermal and magmatic fluids
into coal-bearing strata (Sun et al., 2016; Zheng et al., 2018). Zhang et al. (2023a) identified 15
altered volcanic ashes in this study area, demonstrating the frequent volcanic activity during this
depositional time. And it was confirmed that trace element enrichment in closed coal seams may be
due to the dipping of tonsteins by acid solutions. Correlation analysis of the Hg content of No. 9
coal with trace elements such as Ga, Zr, and Hf revealed a significant correlation (r=0.581-0.705).
Hg enrichment has been widely used as the indicator of volcanic sediment input (Shen et al., 2020),
so the anomalously high values of Hg in the study area may be attributed to the input of Hg
elements due to the frequent volcanic activities at that time, and the Hg in the tonsteins was leached
into the closing coals by leaching of the acid solution.



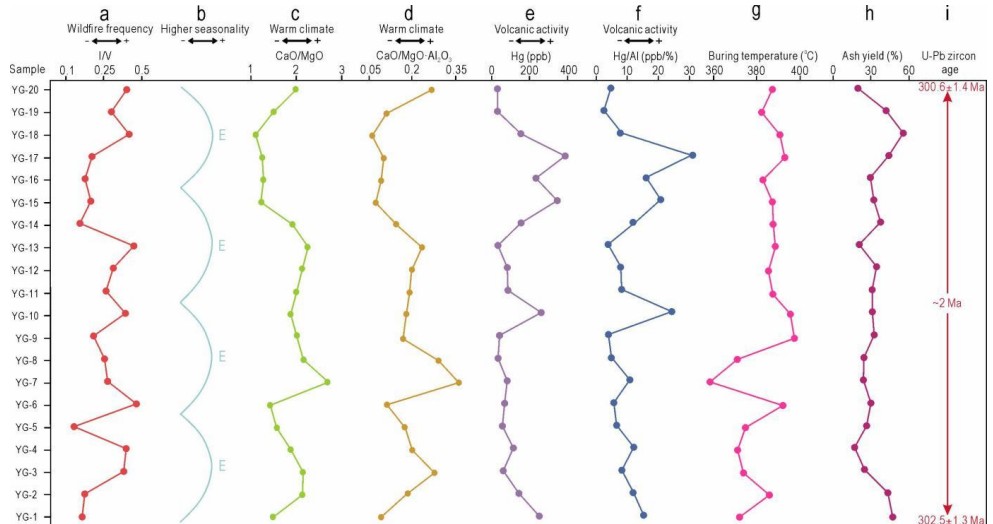

**Fig. 5** Comprehensive analysis map of No.9 coal in Yaogou Mine. (a) Inertinite/ Vitrinite variations in 20 coal

samples. (b) Long eccentricity orbital cycle variation. (c) CaO/MgO trends in 20 coal samples. (d)

CaO/MgO · Al$_2$O$_3$ trends in 20 coal samples. (e) Hg content trends in 20 coal samples. (f) Hg/Al trends in 20 coal

samples. (g) Combustion temperature trends in 20 coal samples. (h) Ash yield trends in 20 coal samples. (i) Age of

No. 9 coal, referred to Zhang et al. (2023a).

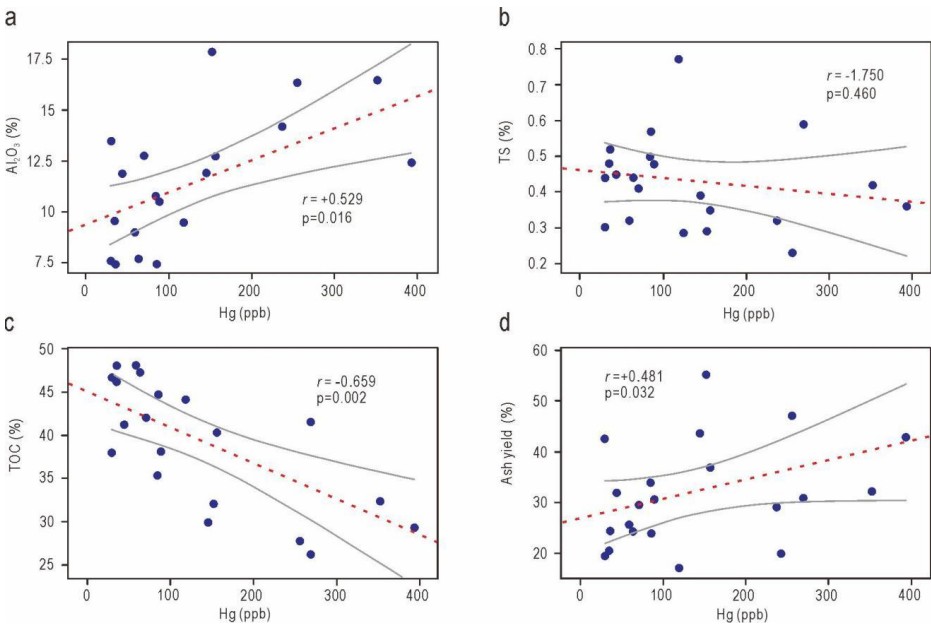

**Fig. 6** (a) Correlation between Hg and Al$_2$O$_3$. (b) Correlation between Hg and TS. (c) Correlation between Hg and



TOC. (d) Correlation between Hg and ash yield. Spearman's P and r values are shown. Red lines denote the
best-fit linear regression line. Gray curve lines denote the 95% confidence interval.

### 5.3 Gzhelian wildfires and climate change

### 5.3.1 Regional wildfire trends

Wildfire activity during geologic history has been influenced primarily by changes in
atmospheric oxygen levels, climate and vegetation (Belcher and Hudspith, 2017; Baker, 2022). For
example, Glasspool and Scott (2010) suggested that high levels of oxygen in the atmosphere
influence increased wildfire activity on different time scales. Baker (2022) found that past periods
of global climate change on Earth have been accompanied by increased wildfire activity. Vegetation
type changes also influence wildfire occurrence, for example, the rise of angiosperms during the
Cretaceous may have led to an increase in wildfire activity at that time (Lü et al., 2024).
The level of oxygen in the atmosphere plays a key role in the occurrence of wildfires (Scott,
2000); when oxygen levels are below 15% it will not be possible to sustain vegetation burning, and
when levels are above 23% humid vegetation will continue to burn. Based on model predictions, the
oxygen level of 24.6% in the Gzhelian is higher than current oxygen levels and can sustain wildfire
burning in humidter environments (Fig. 7).
CaO/MgO can be used as an indicator for paleoclimate reconstruction (Chen and Wan, 1999;
Sun et al., 2010; Liu et al., 2013; Fu et al., 2018), and the ratio of $CaO/MgO \cdot Al_2O_3$ can more
sensitively respond to the temperature changes at that time, with high values indicated for warm
periods, and low values indicated for cold periods (Chen and Wan, 1999; Fu et al., 2018; Zhang et
al., 2020b; Zhang et al., 2021). In this study, periods of high wildfire frequency (e.g., YG-10, YG-11,
YG-12, YG-13) had higher values of CaO/MgO and $CaO/MgO \cdot Al_2O_3$, indicated of a warm and
humid climate (Fig. 5). Periods of lower wildfire frequency (e.g., YG-14, YG-15, YG-16, YG-17)



had lower CaO/MgO and CaO/MgO · $Al_2O_3$ values, indicated a cold and dry climate (Fig. 5).
Although wildfires were more easily caused when the climate was dry, cold climates with lower
temperatures were not favourable for vegetation development (Brovkin, 2002), and a dryer climate
meant less rainfall, both of which led to a decrease in fuels, hence a less frequent occurrence of
wildfires. However, in warm and humid climatic periods, where seasonal variations were stronger,
sufficient rainfall and suitable temperatures promote the development of vegetation, leading to a
significant accumulation of fuels, hence more favourable for wildfires to occur (Litt et al., 2014;
Stockhecke et al., 2016; Swain, 2021). Furthermore, previous research has found that high
frequencies of wildfire activity have occurred in moderately humid environments (Daniau et al,
2012; Marlon et al, 2013). This study predicts oxygen levels sufficient to sustain wildfires in humid
conditions for sustained burning, so sufficient fuel accumulation may have been a major factor in
the frequency of wildfires during this period.
Many scholars have examined charcoal abundance changes in Quaternary sediments to show
that orbital cycle forcing climate drives wildfire activity (Verardo and Ruddiman, 1996; Thevenon
et al., 2004; Zhou et al., 2007; Daniau et al., 2013, 2023; Kappenberg et al., 2019; Hao et al., 2020;
Shi et al., 2020; Zhang et al., 2023b). Zhou et al. (2023), based on a high-resolution charcoal
fragment record from the Wushan Basin of the Tibetan Plateau during the Middle Miocene, found
that orbital period (short eccentricity, slope, and age difference) forced climate change to control
wildfire activity by influencing the amount of vegetative fuels and their combustibility. In this study,
Coal 9 has a floor age of $300.6 \pm 1.4$ Ma, a roof age of $302.5 \pm 1.3$ Ma, and a depositional age of
~1.9 Ma (Zhang et al., 2023a). We explored the effect of the orbital cycle on wildfires using long
eccentricity. As shown in Fig. 5(b-d), the variation of the long eccentricity had a good correlation
with our climatic indicators (CaO/MgO, CaO/MgO · $Al_2O_3$). The maximum eccentricity is





associated with a warmer and more humid period, while the minimum eccentricity is associated
with a colder and dryer period. When the orbital cycle eccentricity is at maximum, the sun was at
perihelion and the amount of insolation was relatively abundant, leading to a sufficient
accumulation of fuels, which in turn led to an increase in wildfire activity (Kappenberg et al., 2019;
Hollaar et al., 2021; Qiu et al., 2023; Zhang et al., 2023b). This is consistent with the study by
Kappenberg et al. (2019) who found that orbital forcing leads to peak wildfire activity during warm
and humid periods.
Combined with the above research, the factors controlling wildfires in this study area are
shown in Fig. 8. When the eccentricity was at a minimum, the Earth received insufficient insolation,
the climate was cold and dry, and seasonal variations were not evident (Hollaar et al., 2021; Huang
et al., 2024). The lack of suitable environments for survival at that time resulted the low amount of
vegetation. There may have been a low frequency of wildfires occurring at this stage, which
produced less charcoal preserved in the peatland (Fig. 8a). When the eccentricity was at a maximum,
the Earth received sufficient insolation, the climate was warm and humid, and seasonal variations
were evident (Hollaar et al., 2021; Huang et al., 2024). Suitable environments lead to the
proliferation of vegetation, which in turn accumulate large amounts of fuel. More rainfall was also
followed by more lightning, leading to increased chances of ignition. Thus there were more frequent
wildfires during the period, which produced more charcoal preserved in the peatlands (Fig. 8b). In
this study, intensive wildfires did not correlate to peaks in Hg concentrations, and the anomalous
enrichment of Hg was most likely due to the frequent volcanic activity occurring at that time. When
the volcano erupts, the ash can carry large Hg transport (Coufalik et al., 2018). When volcanic ash
passes through peatlands, the Hg it carried naturally settles and the strong adsorption of Hg by
peatlands leads to anomalous enrichment of Hg at that time (Fig. 8a) (Yudovich and Ketris, 2005).





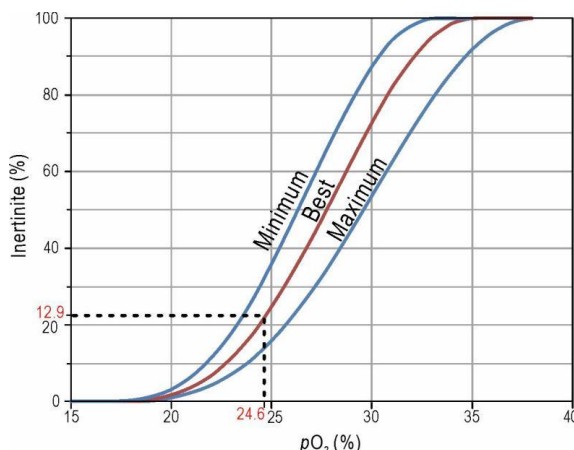


**Fig. 7** The atmospheric oxygen concentration during the Late Carboniferous Taiyuan Formation in the Ordos
Basin based on the prediction model proposed by Glasspool et al. (2015).

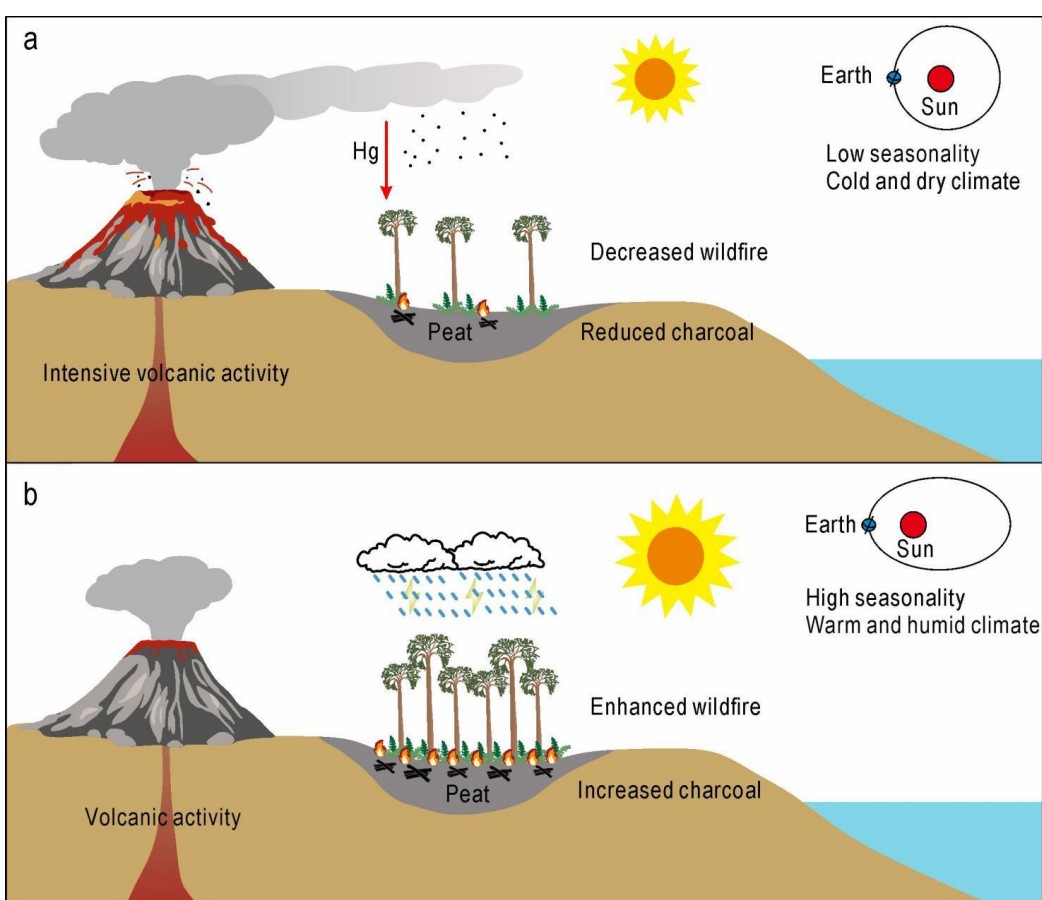


**Fig. 8** Illusrtation of wildfires activity forced by the long eccentricity orbital cycle in the study area.



### 5.3.2 The geographical distribution of published evidence in the Gzhelian


The distribution of wildfires on Earth is not random, and the spatial distribution of wildfires in
the modern world is affected by vegetation, climate, topography, and human activities (Krawchuk et
al., 2009). Studies have shown that frequent wildfires in modern environments mainly occur in
tropical forests and savannas (Mouillot and Field, 2005; Giglio et al., 2006; Flannigan et al., 2009,
2013). In this study, based on the distribution of climate zones published by Boucot et al. (2013), it
was found that all of the wildfire records for the Gzhelian occurred in the tropical climate zone (Fig.
9). It was similar to the modern distribution of wildfires. In the tropics of the Gzhelian, different
biomes are distributed, mainly divided into humid to semi-humid wetland groups and semi-humid
to semi-arid wetland groups, and the peatlands were under alternating humid and dry changing
environments (DiMichele et al., 2010; DiMichele, 2014). In warm and humid environments,
vegetation grows densely and generates large fuel accumulations. And in arid environments, fuel
burnability is enhanced, which in turn promotes more frequent wildfires (Denis et al., 2017; Baker,
2022). Hence the frequent occurrence of wildfires in the tropics with alternating seasons.
In view of the spatial distribution, the Gzhelian wildfire records were all located at low
latitudes near the equator. In the Gzhelian, the southern part of the Gondwana was covered by a
large ice sheet, resulting in the lack of vegetation and the lack of fuel needed for wildfires to burn
(Scotese, 2021) (Fig. 9). Similarly in the northern part of the Gondwana and the northern part of the
Laurasia there were extensive arid climate zones spread across the mainland that lacked the fuels
needed for wildfire burning, resulting in the absence of wildfire records (Fig. 9). However, it is
worth noting that widespread biases, such as taphonomy, research preferences, and sampling bias,
may result in global wildfire data for the Gzhelian not representing the true situation at that time
(Brown et al., 2012; Hamad et al., 2012; Lu et al., 2021; Lü et al., 2024). For example, this study





432 found no evidence of wildfires that have been reported in warm temperate regions (Fig. 9).

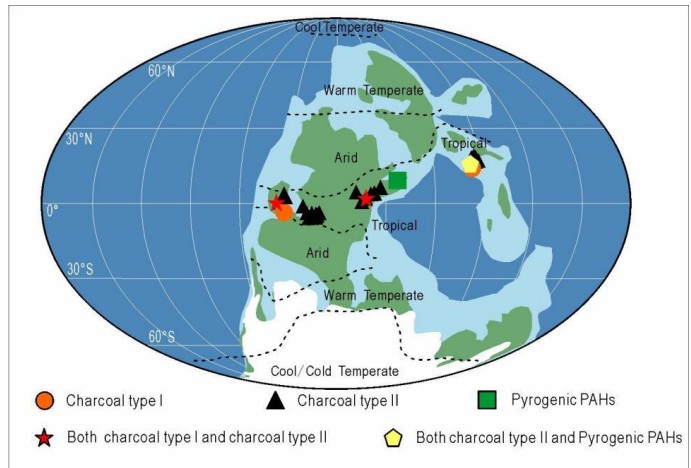


434 **Fig. 9** Paleogeographic distribution of published wildfire occurrences during the Gzhelian. Plate reconstructions

435 from Scotese (2016) with paleoclimatic zones based on Boucot et al. (2013).

## 6. Conclusion

437  In this study, the detailed analysis of the inertinite in the No. 9 coal of the Yaogou coal mine in

438 the Ordos Basin, China, demonstrated that frequent wildfires had occurred in the region during the

439 Gzhelian of the Late Carboniferous and were mainly low-temperature fires. The frequency of

440 wildfires during this period may be related to the forcing of the long eccentricity orbital cycle,

441 where changes in the climate due to changes in the orbital cycle affect the fuel load. The review of

442 the global wildfire record of the Gzhelian that wildfires were concentrated in low-latitude tropical

443 climates, and the distribution of wildfires may have been limited by the climate and fuels available

444 at that time. The anomalous enrichment of Hg in No. 9 coal was probably due to inputs of Hg from

445 the frequent volcanic activity occurring at that time, and was not related to wildfire burning. The

446 different types of wildfire burning may limit the release and enrichment of Hg.

## Author Contribution

448  All the authors have actively participated in the preparation of the manuscript. Wenxu Du was



responsible for data curation, methodology, and writing; Dawei Lv was responsible for funding
acquisition, project administration, and supervision; Zhihui Zhang was responsible for funding
acquisition and methodology; Munira Raji was responsible for review; Cuiyu Song was responsible
for conceptualization and methodology; Luojing Wang was responsible for visualization and
investigation; Zekuan Li was responsible for investigation; Kai Cao was responsible for software
and validation; Ruoxiang Yuan was responsible for visualization and investigation; Yuzhuang Sun
was responsible for review. All authors read and approved the final proof.
**Competing Interests**
The authors declare that they have no conflict of interest.
**Acknowledgement**
This work was financially supported by the National Natural Science Foundation of China
(Grant No. 42102127) and the Natural Science Foundation of Shandong Province (Grant No.
ZR2021QD087), Shandong University of Science and Technology (Grant No. 2018TDJH101). We
think the Deep-Time Digital Earth program (DDE) to support this work. We thank Jincheng Xu to
helped with this study. We gratefully acknowledge Professor James C. Hower for his valuable
comments on this paper.

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
