# Peer review of "Orbital-scale climate dynamics impacts on Gzhelian peatland wildfire activity in"

_Climate of the Past, 2024_

## Referee Comment (RC1)

Principal criteria: Good

**General comments**

Thank you for the opportunity to review this work. This manuscript represents an interesting contribution within the scope of Climate of the Past, applying a novel combination of geochemical methods to a coal deposit in the the Ordos Basin (North China Craton) to interpret the frequency of Gzhelian wildfires in low latitude Pangea, and hypothesize about the potential orbital/climatic drivers. These questions are of broad interest to the paleoclimate community, especially those of us interested in deconvolving the complex drivers of the Earth's climate system evolution in the Late Paleozoic world. The scientific approach is clearly outlined, the combination of methods applied are creative/complementary, and the results are generally presented in a clean and well-structured way. However, I am not convinced about the relationship between wildfire frequency and the (CaO/MgO)*Al ratios, and I have some concerns about making interpretations about paleo-temperatures/humidities based on these proxies (specifics below). Additionally, I think it's acceptable to interpret loosely that wildfire frequency "may be related to the forcing of the long eccentricity orbital cycle", but to interpret this with significance – and include "orbital scale" in the title may require a more quantitative astrochronologic model.

**Specific comments**

- The methods are missing for how the authors "explored the effect of the orbital cycle on wildfires using long eccentricity" – Where are the values for long eccentricity orbital cycle variation (Fig. 5) derived from? And how was the relationship explored between long eccentricity orbital cycle variation and the data obtained in this study? It appears that the interpretation that orbital forcing is driving wildfire frequency is simply a visual observation (Fig. 5). Based on this, I think it is acceptable to interpret that wildfire frequency "may be related to the forcing of the long eccentricity orbital cycle" but to interpret this with significance – and include "orbital scale" as the beginning of the title requires a more quantitative astrochronologic model (comparing curves from Fig. 5 to the known long eccentricity orbital cycle variation from the Late Carboniferous).
- In this study, high (Ca/Mg)*Al during periods of greater wildfire frequency (YG #10-13) is interpreted to reflect a warm and humid climate, and low (Ca/Mg)*Al during periods of less frequent wildfires (YG #14-17) is interpreted to reflect a cold and dry climate. There are a few issues with this: (1) The relationship between I/V and (Ca/Mg)*Al (Fig. 5) even visually does not appear to be consistent. To make this point requires a crossplot with a trendline (and r2 value) showing this relationship rather than just listing YG #10-13 as high/hot humid

and YG #14-17 as low/cold arid. Also (2) please consider that Ca and Al content can be influenced by changes in provenance and/or transport mechanisms as opposed to recording a signal of climate (cf., Demirel-Floyd et al., 2023; https://doi.org/10.1130/B36888.1). At first, this seems like it may not be as much of an issue given that these are coal deposits, but when I look at Figure 5 next to Figure 1, I notice a lot of variability in the Ca/Mg ratio within the same coal beds… It also in some places (e.g., the base YG #1-3 and top YG #19-20 of the section) it seems like the mineralogy (Al, Ca) may be responding to the influx of ash. Accordingly, it may be helpful to have a "j" column on Fig. 5 with the same red/black symbology from Fig. 1 to demonstrate which of these samples are from the same contiguous coal beds (separated by detrital influx) and where there are ash beds dispersed between them.

- There are several tonsteins directly within the study area that demonstrate the abundance of volcanic activity during this time… It might be worth mentioning that globally during the Late Carboniferous there is evidence of anomalously frequent volcanism (see Soreghan et al., 2019) that complements the data presented herein and further supports the interpretation that Hg enrichment is driven more by volcanism than by wildfires during this time.

- It is not critical, but dating the interbedded tonsteins would strengthen the story about wildfire frequency derived from I/V ratios, and an astrochronologic model (if the authors decide to go in that direction).

- It might be interesting to evolve the discussion in 5.3.2 to explicitly state how the (new) data fits into this regional/global framework, and also to acknowledge that the No. 9 coal of the Yaogou coal mine only records 2 Myr of climatic variation—It will be interesting to continue to test how these interpretations fit within the broader Gzhelian record/broader regional Gzhelian paleoclimate interpretations.

**Technical corrections**

- I suggest highlighting on Fig. 1 which of the tonsteins in the No. 9 coal seam constrain the age and maybe list the dates as well (from *Zhang et al. 2023a?)*
- Missing a % sign on Line 181
- In Table 1 the column titles maybe make a line break to differentiate what is in which category (organic vs. inertinite macerals)
- In Table 1 is "total minerals" the ash yield?
- Consider adding calculated V/I and Ro from each sample in either Tables 1 or 2 for easy reference.

---

## Referee Comment (RC2)

[referee-annotated manuscript omitted]

---

## Author Comment (AC1)

**Point-by-point responses (in blue) to the Editor and Reviewers' comments:**

Manuscript No.: cp-2024-42

**Reviewer #1:**

1. The methods are missing for how the authors "explored the effect of the orbital cycle on wildfires using long eccentricity" - Where are the values for long eccentricity orbital cycle variation (Fig. 5) derived from? And how was the relationship explored between long eccentricity orbital cycle variation and the data obtained in this study? It appears that the interpretation that orbital forcing is driving wildfire frequency is simply a visual observation (Fig. 5). Based on this, I think it is acceptable to interpret that wildfire frequency "may be related to the forcing of the long eccentricity orbital cycle" but to interpret this with significance - and include "orbital scale" as the beginning of the title requires a more quantitative astrochronologic model (comparing curves from Fig. 5 to the known long eccentricity orbital cycle variation from the Late Carboniferous).

Thanks for your suggestions. We have revised the wording to 'may be related to the forcing of the long eccentricity orbital cycle.' Due to the limited number of samples, we were unable to establish a quantitative astronomical orbital model. However, following your advice, we found an appropriate curve for the long eccentricity orbital cycle variations. Wu et al. (2023) reviewed Paleozoic cyclostratigraphy and established an astronomical time scale based on published paleoclimate proxy time series. By comparing our data with the quantitative astronomical orbital model under age constraints, we found that the results were consistent with our initial assumptions. Additionally, we are merely proposing a hypothesis, and further research will be conducted in this area in future studies. We also hope to spark interest among other scholars in studying the influence of Late Paleozoic orbital cycles on wildfire intensity.

[Figure]

**Fig. 5** Comprehensive analysis map of No.9 coal in Yaogou Mine. (a) Age of No. 9 coal, referred to Zhang et al. (2023a). (b) Inertinite/ Vitrinite variations in 20 coal samples. (c) Long eccentricity orbital cycle variation, referred to Wu et al. (2023). (d) CaO/MgO trends in 20 coal samples. (e) CaO/MgO • Al2O3 trends in 20 coal samples. (f) Hg content trends in 20 coal samples. (g) Hg/Al trends in 20 coal samples. (h) Combustion temperature trends in 20 coal samples. (i) Ash yield trends in 20 coal samples.

2. In this study, high (Ca/Mg)*Al during periods of greater wildfire frequency (YG #10-13) is interpreted to reflect a warm and humid climate, and low (Ca/Mg)*Al during periods of less frequent wildfires (YG #14-17) is interpreted to reflect a cold and dry climate. There are a few issues with this: (1) The relationship between I/V and (Ca/Mg)*Al (Fig. 5) even visually does not appear to be consistent. To make this point requires a crossplot with a trendline (and r2 value) showing this relationship rather than just listing YG #10-13 as high/hot humid and YG #14-17 as low/cold arid. Also (2) please consider that Ca and Al content can be influenced by changes in provenance and/or transport mechanisms as opposed to recording a signal of climate (cf., Demirel-Floyd et al., 2023; https://doi.org/10.1130/B36888.1). At first, this seems like it may not be as much of an issue given that these are coal deposits, but when I look at Figure 5 next to Figure 1, I notice a lot of variability in the Ca/Mg ratio within the same coal beds⋯ It also in some places (e.g., the base YG #1-3 and top YG #19-20 of the section) it seems like the mineralogy (Al, Ca) may be responding to the influx of ash. Accordingly, it may be helpful to have a "j" column on Fig. 5 with the same red/black symbology from Fig. 1 to demonstrate which of these samples are from the same contiguous coal beds (separated by detrital influx) and where there are ash beds dispersed between them.

Thanks for your suggestions. We attempted a correlation analysis between I/V and (Ca/Mg)*Al, but the results were not ideal. As you mentioned, the concentrations of calcium and aluminum may be influenced by changes in source material and/or transport mechanisms. Zhang et al. (2023) already noted that there were frequent and intense volcanic activities in Coal Seam No. 9, and a certain amount of volcanic clastic material was present in the coal. Some of the Mg and Ca may have been affected by volcanic activity, leading to signal interference. As a result, the outcomes of the quantitative analysis were not ideal, and we were only able to conduct qualitative discussions.

3. There are several tonsteins directly within the study area that demonstrate the abundance of volcanic activity during this time⋯ It might be worth mentioning that globally during the Late

Carboniferous there is evidence of anomalously frequent volcanism (see Soreghan et al., 2019) that complements the data presented herein and further supports the interpretation that Hg enrichment is driven more by volcanism than by wildfires during this time.

Thank you very much for your suggestion. The relevant content has been added to the manuscript.

4. It is not critical, but dating the interbedded tonsteins would strengthen the story about wildfire frequency derived from I/V ratios, and an astrochronologic model (if the authors decide to go in that direction).

Thanks for your valuable review. We appreciate your suggestion to date the interbedded tonsteins to strengthen our story. However, due to the focus of our current study and resource constraints, we have decided not to pursue this additional analysis.

5. It might be interesting to evolve the discussion in 5.3.2 to explicitly state how the (new) data fits into this regional/global framework, and also to acknowledge that the No. 9 coal of the Yaogou coal mine only records 2 Myr of climatic variation-It will be interesting to continue to test how these interpretations fit within the broader Gzhelian record/broader regional Gzhelian paleoclimate interpretations.

Thanks for your suggestions. The relevant content has been revised.

6. I suggest highlighting on Fig. 1 which of the tonsteins in the No. 9 coal seam constrain the age and maybe list the dates as well (from Zhang et al. 2023a?)

Thanks and done.

7. Missing a % sign on Line 181

Thanks and done.

8. In Table 1 the column titles maybe make a line break to differentiate what is in which category (organic vs. inertinite macerals)

Thanks and done.

9. In Table 1 is "total minerals" the ash yield?

Thanks for your suggestion. "Total minerals" is not ash yield. "Total minerals" indicates the inorganic minerals present in the coal sample.

10. Consider adding calculated V/I and Ro from each sample in either Tables 1 or 2 for easy reference.

Thanks for your suggestion. The relevant data have been added to Table 2.

**Table 2**

The I/V, $R_o$ and chemical element data of 20 coal samples from the No. 9 coal seam of the Yaogou coal mine in the Ordos Basin. The $Al_2O_3$, TS, and TOC were from Zhang et al. (2023a).

| Sample | I/V | $R_o$ (%) | Hg (ppb) | $Al_2O_3$ (%) | TS (%) | TOC (%) |
|--------|------|-----------|----------|---------------|--------|---------|
| YG-1   | 0.18 | 1.59      | 255      | 16.33         | 0.23   | 27.82   |
| YG-2   | 0.2  | 1.71      | 145      | 11.94         | 0.39   | 29.92   |
| YG-3   | 0.4  | 1.61      | 63.2     | 7.67          | 0.44   | 47.30   |
| YG-4   | 0.41 | 1.59      | 118      | 9.46          | 0.77   | 44.14   |
| YG-5   | 0.14 | 1.62      | 58.6     | 8.96          | 0.32   | 48.07   |
| YG-6   | 0.47 | 1.77      | 70.4     | 12.75         | 0.41   | 42.04   |
| YG-7   | 0.31 | 1.48      | 85.1     | 7.44          | 0.57   | 44.72   |
| YG-8   | 0.3  | 1.59      | 35.4     | 7.42          | 0.52   | 48.04   |
| YG-9   | 0.24 | 1.81      | 43.5     | 11.87         | 0.45   | 41.17   |
| YG-10  | 0.42 | 1.8       | 269      | 10.5          | 0.59   | 41.51   |
| YG-11  | 0.31 | 1.73      | 88.5     | 10.53         | 0.48   | 38.10   |
| YG-12  | 0.35 | 1.71      | 83.9     | 10.74         | 0.5    | 35.33   |
| YG-13  | 0.46 | 1.74      | 34.5     | 9.57          | 0.48   | 46.26   |
| YG-14  | 0.17 | 1.73      | 156      | 12.77         | 0.35   | 40.37   |
| YG-15  | 0.23 | 1.72      | 352      | 16.48         | 0.42   | 32.35   |
| YG-16  | 0.2  | 1.69      | 237      | 14.23         | 0.32   | 36.72   |
| YG-17  | 0.24 | 1.77      | 393      | 12.42         | 0.36   | 29.26   |
| YG-18  | 0.43 | 1.75      | 152      | 17.88         | 0.29   | 32.05   |
| YG-19  | 0.34 | 1.68      | 29.5     | 13.53         | 0.3    | 37.95   |
| YG-20  | 0.42 | 1.72      | 29.6     | 7.57          | 0.44   | 46.68   |

**Reference**

Wu, H., Fang, Q., Hinnov, L. A., Zhang, S., Yang, T., Shi, M., and Li, H.: Astronomical time scale for the Paleozoic Era. Earth-Science Reviews, 104510, https://doi.org/10.1016/j.earscirev.2023.104510, 2023.

Zhang, Z., Lv, D., Hower, J. C., Wang, L., Shen, Y., Zhang, A., Xu, J., and Gao, J.: Geochronology, mineralogy, and geochemistry of tonsteins from the Pennsylvanian Taiyuan Formation of the Jungar Coalfield, Ordos Basin, North China, International Journal of Coal Geology, 267, 104183, https://doi.org/10.1016/j.coal.2023.104183, 2023.

---

## Author Comment (AC2)

**Point-by-point responses (in blue) to the Editor and Reviewers' comments:**

Manuscript No.: cp-2024-42

**Reviewer #2:**

1. Line 34: published

Thanks and done.

2. Line 35: P

Thanks and done.

3. Line 52: Orbital periods

Thanks and done.

4. Line 52: Whether

Thanks and done.

5. Line 124: provide a better understanding

Thanks and done.

6. Line 128: maceral composition

Thanks and done.

7. Line 129: Vitrinite reflectance

Thanks and done.

8. Line 133: by

Thanks and done.

9. Line 135: and

Thanks and done.

10. Line 143: also

Thanks and done.

11. Line 145: unreliable

Thanks and done.

12. Line 150: two

Thanks and done.

13. Line 151: also

Thanks and done.

14. Line 168: Coal maceral data, reported on a mineral matter free basis (mmf),

Thanks and done.

15. Line 169: Vitrinite

Thanks and done.

16. Line 169: in all samples

Thanks and done.

17. Line 171: Liptinite contents were low

Thanks and done.

18. Line 175: (Table 1)

Thanks and done.

19. Line 175: The

Thanks and done.

20. Line 176: and

Thanks and done.

21. Line 180: VRo

Thanks and done.

22. Line 187: coefficient

Thanks and done.

23. Line 187: I would not consider correlation coefficient of 0.529 to be "highly" significant.

Thanks for your suggestions. We have removed the description of 'highly'.

24. Line 188: a

Thanks and done.

25. Line 191: a

Thanks and done.

26. Line 209: images

Thanks and done.

27. Line 216: In addition,

Thanks and done.

28. Line 220: vacuoles

Thanks and done.

29. Line 227: represent

Thanks and done.

30. Line 240: represent

Thanks and done.

31. Line 243: have

Thanks and done.

32. Line 244: are likely

Thanks and done.

33. Line 247: thermally immature

Thanks and done.

34. Line 248: have

Thanks and done.

35. Line 249: at least

Thanks and done.

36. Line 253: This is actually a wide range of variability.

Thanks for your suggestion. The revisions have been completed.

37. Line 258: characteristics

Thanks and done.

38. Line 266: This is a wide range of variability

Thanks for your suggestion. The revisions have been completed.

39. Line 271: How do you know these are from fern tissues?

Thank you for your suggestion. I have added additional explanations regarding ferns and included the relevant references.

40. Line 297: does not correlate with

Thanks and done.

41. Line 320: What is a "closed" coal seam?

Thanks. It has been corrected.

42. Line 321: leaching

Thanks and done.

43. Line 325: activity

Thanks and done.

44. Line 326: emplaced

Thanks and done.

45. Line 342: influence

Thanks and done.

46. Line 353: is sensitive

Thanks and done.

47. Line 353: indicating

Thanks and done.

48. Line 354: indicating

Thanks and done.

49. Line 356: which are indicative of

Thanks and done.

50. Line 358: indicating

Thanks and done.

51. Line 360: drier

Thanks and done.

52. Line 361: and

Thanks and done.

53. Line 377: The peat literature suggests that peat accumulation events are much younger, typically between 8,000 and 10,000 years.

Thank you for your suggestion. The No. 9 coal contains multiple layers of volcanic ash formed by volcanic activity (see Fig. 1), so 1.9 Ma represents the sedimentation age including both peat and volcanic ash.

54. Line 380: Maximum

Thanks and done.

55. Line 383: with

Thanks and done.

56. Line 383: levels

Thanks and done.

57. Line 391: in reduced

Thanks and done.

58. Line 397: ,

Thanks and done.

59. Line 401: volcanoes erupt

Thanks and done.

60. Line 401: facilitate

Thanks and done.

61. Line 402: falls out on

Thanks and done.

62. Line 416: This is

Thanks and done.

63. Line 420: proliferates

Thanks and done.

64. Line 420: In

Thanks and done.

65. Line 422: frequency of

Thanks and done.

66. Line 422: is a function of

Thanks and done.

67. Line 423: Spatially,

Thanks and done.

68. Line 430: uncertain

Thanks and done.

69. Line 430: not representing the true situation at that time

Thanks and done.

70. Line: had

Thanks and done.

71. Line: of the Late Carboniferous

Thanks and done.

---

## Author Comment (AC3)

**Point-by-point responses (in blue) to the Editor and Reviewers' comments:**

Manuscript No.: cp-2024-42

**Title:** Orbital-scale climate dynamics may impact Gzhelian peatland wildfire activity in the Ordos Basin

**Editor:**

1. Line 110: what is a "<20 top size"? What are the units here?

Thanks for your suggestion. It was an error in our work, and the unit has been added.

2. Line 141-142: To increase the comprehensive of the database, it would be good to also use "Virgilian wildfire" to capture North American incidences.

Thanks for your suggestion. We have expanded the keyword search to ensure the accuracy of the data.

3. Line 162: should be "Scotese"

Thanks and done.

4. Line 167: Could a principal component analysis (PCA) to see how all of these variables co-relate shed additional insight?

Thanks for your suggestions. We attempted to perform principal component analysis (PCA), but the results were not ideal and did not provide new insights for our research.

5. Line 201: Table 1— fix such that column labels are not truncated

Thanks and done.

**Table 1**

The coal micro component contents of 20 samples of No. 9 coal seam from Yaogou Mine in Ordos Basin

| Sample No. | Percentage of the total organic macerals (vol.%) | | | Percentage of the total inertinite macerals (vol.%) | | | | | TOM (vol.%) | Total minerals (vol.%) |
|---|---|---|---|---|---|---|---|---|---|---|
| | Vitrinite | Inertinite | Liptinite | Fusinite | Semifusinite | Macrinite | Micrinite | Inertodetrinite | | |
| YG-1 | 79.6 | 14.6 | 5.8 | 1.1 | 9.3 | 0 | 0.2 | 4 | 53.9 | 46.1 |
| YG-2 | 80.5 | 16 | 3.5 | 2.4 | 10.6 | 0.6 | 0.6 | 1.8 | 60.4 | 39.6 |
| YG-3 | 68.3 | 27.4 | 4.3 | 0.9 | 14.1 | 1.5 | 2.6 | 8.3 | 81.2 | 18.8 |
| YG-4 | 64.5 | 26.8 | 8.7 | 0.4 | 13.3 | 2.4 | 0.4 | 10.3 | 84.8 | 15.2 |
| YG-5 | 84.7 | 11.9 | 3.4 | 0.4 | 7.4 | 1.5 | 0.2 | 2.4 | 84.9 | 15.1 |
| YG-6 | 65.2 | 30.8 | 4 | 0 | 14.6 | 2.4 | 0.4 | 13.4 | 86.2 | 13.8 |
| YG-7 | 72.2 | 23 | 4.8 | 3 | 10.7 | 3 | 0.8 | 5.5 | 87.1 | 12.9 |
| YG-8 | 70.8 | 21.5 | 7.7 | 2.5 | 12.2 | 1.5 | 0.2 | 5.1 | 81.8 | 18.2 |
| YG-9 | 78.4 | 19.1 | 2.5 | 0 | 9.8 | 2.5 | 0.4 | 6.4 | 80.1 | 19.9 |
| YG-10 | 66.1 | 27.4 | 6.5 | 3.8 | 13.1 | 3.2 | 0.6 | 6.7 | 83.9 | 16.1 |
| YG-11 | 75.2 | 23.5 | 1.3 | 0.6 | 12 | 3.4 | 0.2 | 7.3 | 79.6 | 20.4 |
| YG-12 | 72.1 | 25.1 | 2.8 | 0.9 | 15.4 | 2.2 | 0.4 | 6.2 | 73.9 | 26.1 |
| YG-13 | 65.8 | 30.2 | 4 | 0.8 | 17.2 | 4.1 | 1.8 | 6.3 | 86.2 | 13.8 |
| YG-14 | 82 | 14.1 | 3.9 | 0 | 8.3 | 2.1 | 0 | 3.7 | 81.3 | 18.7 |
| YG-15 | 80.2 | 18.4 | 1.4 | 0 | 11.2 | 0.8 | 0.6 | 5.8 | 77 | 23 |
| YG-16 | 81.8 | 16.5 | 1.7 | 0 | 10.4 | 1.2 | 1.2 | 3.7 | 85.2 | 14.8 |
| YG-17 | 78.1 | 18.4 | 3.5 | 2.7 | 9 | 1.7 | 0.5 | 4.5 | 59.5 | 40.5 |
| YG-18 | 66.8 | 28.9 | 4.3 | 0.8 | 11.3 | 4.7 | 0 | 12.1 | 59.9 | 40.1 |
| YG-19 | 73 | 24.9 | 2.1 | 1.1 | 8.9 | 2.9 | 0.4 | 11.6 | 71.5 | 28.5 |
| YG-20 | 65 | 27.2 | 7.8 | 3.2 | 11 | 4.1 | 2 | 6.9 | 81.8 | 18.2 |

6. Line 214: typo; need extra space

Thanks and done.

7. Line 223: The "Virgilian" is the (approximate) time equivalent to the Gzhelian and was/is widely used in North America, so if you want this to be a more comprehensive database consider using that search term as well.

Thanks for your suggestion. We added this keyword to the search, but unfortunately, no additional data was found.

8. Line 285: noun needed ("organic and inorganic" what?)— maybe "matter"?

Thanks for your suggestion. This sentence has been rewritten.

9. Line 302-303: But, consider the volume of vegetation that is required to result in a given volume of peat (or - even more so - coal). Clearly there will be a greater concentration in the latter owing to this volume difference. Is that the point you are trying to make? I'm finding the reasoning in this paragraph difficult to follow; I think I understand your point, but it is a bit muddled in my reading.

Thanks for your suggestion. This sentence has been rewritten.

10. Line 320-321: I'm not sure what this means— the "…dipping of tonsteins by acid solutions." Are you referring to the leaching of tonsteins by organic acids released/related to the coals?

Sorry, it was a mistake of expression on our part. As you correctly pointed out, our intended meaning was that trace element enrichment in the enclosed coal seams is likely due to leaching by acidic solutions. The statement has been rewritten.

11. Line 321-322: again, PCA might help?

Thanks for your suggestions. We attempted PCA, but the results did not help us improve our analysis.

12. Line 326: unclear what is meant by "closing coals"

Thanks, it has been modified.

13. Fig 5 - it would be easier to analyze the possible correlations by conducting multicomponent statistical analyses. Also, on this plot, each dot represents a coal sample from a correlative coal seam? Ie, are the labels on the left - "YG" - are those all discrete coal layers? Could you add a

graphical stratigraphic log to help visualization of this?

Thanks for your suggestions. However, as Zhang et al. (2023) noted, there were frequent and intense volcanic activities in Coal Seam No. 9, and a certain amount of volcanic clastic material was present in the coal. The clastic material from volcanic activities affects the compound content in the peatland, leading to signal interference. As a result, the outcomes of our quantitative data analysis were not ideal, and we could only conduct qualitative discussions. Based on your suggestions, we have added the graphical stratigraphic log in Fig. 5.

[Figure]

**Fig. 5** Comprehensive analysis map of No.9 coal in Yaogou Mine. (a) Age of No. 9 coal, referred to Zhang et al. (2023a). (b) Inertinite/ Vitrinite variations in 20 coal samples. (c) Long eccentricity orbital cycle variation, referred to Wu et al. (2023). (d) CaO/MgO trends in 20 coal samples. (e) CaO/MgO · Al2O3 trends in 20 coal samples. (f) Hg content trends in 20 coal samples. (g) Hg/Al trends in 20 coal samples. (h) Combustion temperature trends in 20 coal samples. (i) Ash yield trends in 20 coal samples.

14. If I am understanding correctly, you are suggesting that various curves here reflect the eccentricity curve, based on visual comparison. But to really be convincing, I think this would require a more quantitative analysis, and probably with many more data points (than the 20 here).

Thanks for your suggestions. Indeed, with our 20 data points, it is not possible to establish an astronomical orbital cycle. Following Reviewer 1's suggestions, we compared our data again using the known orbital cycle model (Wu et al., 2023) and our age constraints. We found that the results were consistent with our previous assumptions.

15. Line 351: Although such metrics have been used to assess weathering and, by extension, climate, recent studies have called this into question owing to the strong (overriding) control of provenance on these oxides.

Thanks for your suggestions. As you mentioned, these oxides are greatly influenced by the provenance, which introduces interference in our quantitative data analysis. Therefore, we conducted only qualitative analysis to explore the impact of astronomically driven climate change on wildfire activity.

16. Line 360: typo

Thanks and done.

17. Line 377: The 1.9 Ma is not a depositional age, but I think you mean duration here, which means the units should be My. But that's not quite correct, because you have not considered the error bars. Considering the errors on both dates, the duration could be as long as 4.6 My, or as brief as 0.8 My.

Thanks for your suggestion. This sentence has been rewritten.

18. Line 379-380: I do not think that a visual comparison, with so few data points, is sufficient to make this claim.

Thanks for your suggestions. Our work is merely an attempt to propose a hypothesis, and we will further explore this aspect in future studies. We also hope to spark interest among other scholars in the study of the influence of Late Paleozoic orbital cycles on wildfire intensity.

19. Line 382: typo

Thanks and done.

**Reference**

Wu, H., Fang, Q., Hinnov, L. A., Zhang, S., Yang, T., Shi, M., and Li, H.: Astronomical time scale for the Paleozoic Era. Earth-Science Reviews, 104510, https://doi.org/10.1016/j.earscirev.2023.104510, 2023.

Zhang, Z., Lv, D., Hower, J. C., Wang, L., Shen, Y., Zhang, A., Xu, J., and Gao, J.: Geochronology, mineralogy, and geochemistry of tonsteins from the Pennsylvanian Taiyuan Formation of the Jungar Coalfield, Ordos Basin, North China, International Journal of Coal Geology, 267, 104183, https://doi.org/10.1016/j.coal.2023.104183, 2023.